# "I could not bear it": Perceptions of chronic pain among Somali pastoralists in Ethiopia. A qualitative study

Eleonore Baum[1,2,3]*, Sied Abdi[4], Nicole Probst-Hensch[2,3], Jakob Zinsstag[2,3], Birgit Vosseler[5], Rea Tschopp[2,3,6], Peter van Eeuwijk[2,3,7]

1 Institute of Applied Nursing Science IPW, OST - Eastern Switzerland University of Applied Sciences, St. Gallen, Switzerland, 2 Swiss Tropical and Public Health Institute, Allschwil, Switzerland, 3 University of Basel, Basel, Switzerland, 4 School of Nursing and Midwifery, Jigjiga University, Jigjiga, Ethiopia, 5 School of Health Sciences, OST - Eastern Switzerland University of Applied Sciences, St. Gallen, Switzerland, 6 Armauer Hansen Research Institute, Addis Ababa, Ethiopia, 7 Institute of Social Anthropology, University of Basel, Basel, Switzerland

* eleonore.baum@ost.ch

**Data Availability Statement:** The authors confirm that the relevant data for study replication are available within the manuscript (excerpts of transcripts) and its supplementary materials

## Abstract

### Background

Pain is a major public health problem in the Global South, particularly among marginalized communities, such as Somali pastoralists. Yet, the topic of chronic pain has not yet been comprehensively studied in Sub-Saharan Africa, specifically in the Somali region of Ethiopia. Therefore, this study aims to explore the perceptions and notions of chronic pain among Somali pastoralists in this context.

### Methods

This study used an explorative qualitative design. We performed semi-structured, face-to-face interviews with 20 purposively selected female and male Somali pastoralists with chronic pain. For data analysis, we applied the Framework Method by Gale et al. and explained patterns drawing on the Enactive Approach to Pain proposed by Stilwell and Harman.

### Findings

Six different themes emerged: (1) "Pain as a symptom of harsh daily life", (2) "Pain descriptions and dimensions", (3) "Temporality of pain", (4) "Pain-related stigma and stoicism" (5) "Mediating role of spirituality", and (6) "Impact of pain on daily life activities".

### Conclusions

Somali pastoralists described their chronic pain as a multicausal and relational experience. Pastoralists (especially women) commonly refrained from communicating their pain and represented aspects of social stigma and stoicism. The mediating role of spirituality aided pastoralists to make sense of their pain and to ease its impact on their harsh daily life. The

(interview guide). Additional qualitative study data is made available on the Qualitative Data Repository (QDR) (accession number https://doi.org/10.5064/F6N2GOC9).

**Funding:** This 10-year project is part of the Jigjiga University One Health Initiative (JOHI) co-funded by the Swiss Agency for Development and Cooperation (SDC) Project no. 7F-09057.02.01. The project is implemented by Jigjiga University (JJU), Swiss Tropical and Public Health Institute (Swiss TPH) and the Armauer Hansen Research Institute (AHRI). The project is also supported through the doctoral program at OST - Eastern Switzerland University of Applied Sciences and the scholarship program from the Swiss School of Public Health (SSPH+). The funders had no impact on the design, data collection or interpretation of the study results. This study received no additional external funding.

**Competing interests:** The authors have declared that no competing interests exist.

**Abbreviations:** EKNZ, Ethics Committee of Northwest and Central Switzerland; FGC, Female Genital Cutting; IASP, International Association for the Study of Pain; JOHI, Jigjiga One Health Initiative; SRS, Somali Regional State.

findings of this study can contribute to raise awareness of chronic pain issues among pastoralists. They highlight the need for policymakers to prioritize the improvement of pastoralist-specific pain management. Necessary resources and skills should be available within health care facilities. Pain management should be accessible, affordable and culturally acceptable for this population.

## Introduction

Pain is a major global problem often accompanied by severe distress, demoralization and functional impairment. It is associated with poorer quality of life, making pain a major source of physical suffering and economic burden [1, 2]. As a global health priority, treating pain is not receiving the necessary attention [3]. Although pain management is a problem in both Global North and Global South [4], suffering from inadequately and untreated pain is far more worrisome among people living in precarity [5, 6]. Countries with a low human development index (HDI) will experience an increase in cancer cases of 95% until 2040 [7]. Therefore, the burden of cancer-associated pain will be especially troubling in the Global South. Compared to many other essential health interventions that have been recognized as a priority, the need for pain relief has been widely neglected–despite the fact that adequate access to treatment for pain must be ensured by governments according to international human rights law [8].

The International Association of the Study of Pain (IASP) [9] defines pain as an "unpleasant sensory and emotional experience associated with, or resembling that associated with, actual or potential tissue damage" (p. 1). Chronic pain is described as a disease in its own right, as it was recently added to the International Classification of Diseases (ICD-11) [1]. Internationally recognized definitions of pain stem from a Western-oriented scientific approach [10]. However, research shows that pain perception and expression is largely shaped by the person's culture and environment [11–14]. Social values and upbringing have an impact on how people experience, reflect and act on pain [15]. Moreover, marginalized populations, such as Somali pastoralists, remain underrepresented in pain research, making research-based translations into practice for more effective therapeutic measures challenging [10, 16].

## Background

The Somali Regional State (SRS) in Eastern Ethiopia has a population of estimated 5.6 million people. The majority of people are pastoralists or agro-pastoralists depending on livestock and farming for their livelihood [17]. Pastoralists are known to withstand hardships in daily life, such as precarious living and health conditions as well as extreme climates, such as droughts and floods. In the SRS, pastoralists are part of tightly knit social networks within a clan system. Their independence (and distrust) of state-run facilities, such as hospitals, can also be linked to their seasonal mobility and remoteness [18]. Illness and health beliefs of Somali pastoralists, including their views on pain, have been described as pluralistic: Biomedical, traditional, and spiritual health practices are not mutually exclusive and exist in parallel [19].

For pastoralists as a socially and geographically marginalized population, access to biomedical health services is challenging [18, 20]. Studies of the proportion of pastoralists in Ethiopia who use health services to treat their illnesses vary widely according to region, distance to health facilities and other factors such as mobility [21–23]. A study in rural Western Ethiopia found that 36.7% of households reporting illness episodes were self-medicating with modern drugs, most commonly antibiotics and analgesics. The majority of households had consulted

health care professionals [24]. A study in a rural area of Southern Ethiopia revealed that common sources of antimalarial medications were clinics, health posts or health centers (other than malaria control programs) [25].

Common reasons among pastoralists for not visiting health facilities are the far distance of health institutions from the place of residence or cultural beliefs that health services lack efficacy for certain health problems [21]. In addition, the necessary resources especially for treating pain, such as pain medication, are often not available in health care facilities [26]. This diminishes the degree of acceptability for pastoralists. Pastoralists commonly also seek aid from traditional healers or prefer self-treatment when experiencing illness symptoms [22, 27]. In such contexts, pain can lack prioritization as a serious health problem–on the part of pastoralists themselves and also on the part of health professionals [11, 21, 26]. However, Carruth argued that seeking biomedical aid is ever-changing in this population as pastoralists alter their views about healthcare and adapt to new environments [19].

Despite the urgent need for more research, the topic of pain management has not yet been fully studied in Sub-Saharan Africa [12], specifically in Ethiopia [28–30]. However, several studies explored the pain experience of Somali refugees and Somalis living abroad [31, 32]. In a qualitative study with nine Somali women in Sweden, Finnström and Söderhamn found that crying and wailing as an expression of pain is perceived to be a culturally unacceptable behavior. The authors also described that people with a Somali cultural background often rely on informal care and traditional methods of pain relief [32].

Against this background, our study for the first time examines how Somali pastoralists living in Ethiopia perceive and deal with chronic pain.

## Aim

We address the research gap by aiming to gain first insights into pastoralists' perceptions and notions of chronic pain. Therefore, the leading research question at the heart of this article is: How do Somali pastoralists perceive their chronic pain?

This study is part of a larger research program addressing pain management in the Somali Region. As part of the Jigjiga One Health Initiative (JOHI), our study follows a transdisciplinary approach, involving academic and non-academic actors in a transformational research process integrating systemic and practical knowledge [33–35]. JOHI is a research-development project implemented in the Somali Regional State (SRS) in Ethiopia and reaching out to Somaliland and Somalia. It aims to create innovative integrated health systems addressing the health and wellbeing of pastoralist communities.

## Methods

### Study design

To explore sociocultural aspects of pain perception and corresponding experiential and existential domains, we chose an explorative qualitative design guided by the Framework Method [36, 37]. This method seemed particularly suitable for working in multidisciplinary teams with people of different professional and cultural backgrounds. Moreover, this approach to exploring pain perception also provided the possibility of informing both policy and practice [38]. To report this study, we used the Consolidated criteria for reporting qualitative research (COREQ) checklist (S1 Appendix) [39].

## Theoretical framework

The Enactive Approach to Pain proposed by Stilwell and Harman [14] served as the theoretical framework for this study. It complements Engel's biopsychosocial model [40]. The original biopsychosocial model attempts to incorporate the social, psychological and behavioral dimensions of illness into the biomedical disease model. It intends to improve the understanding of a person's perceived illness in a broader manner compared to preceding models representing a strict biological focus [40]. The enactive pain approach goes beyond the biopsychosocial model by doing justice to the lived embodied experience of persons affected and to the importance of their perceived sense of social agency. Positioned at the center of the enactive framework, is the idea of a "field of affordances" [14, 41], a concept first introduced by Gibson in 1979 [42]. Coninx and Stilwell [41] conceptualize pain in terms of "affordances" defined as "perceived possibilities for action" (p. 7837). The enactive approach comprises three central domains: the physiological domain, the phenomenal-existential domain, and the socio-cultural domain. The phenomenal-existential domain refers to the individuals' self-perception of pain and of the world around them.

This framework is especially compatible with our study since it gives room for persons' lived experience of chronic pain and considers the impact of the environment on individual pain experience. Its holistic and integrative nature steers against the methodological risk of reductionist fragmentation and allows to overcome strictly separated disciplinary domains in research and clinical practice. The enactive approach to pain stipulates that the biological, psychological and social domains are continuously conditioned by each other and subject to cross-cutting processes [14, 41].

## Sampling

Eligible were adult pastoralists or agro-pastoralists with chronic pain, as defined by the International Association for the Study of Pain (IASP) [1]. Pastoralists with cognitive impairment and/or communication difficulties, as well severely ill patients (determined by the responsible health professionals) were excluded from our study. Initially, we identified participants by using a purposive sampling strategy aiming at maximum variability (e.g., concerning age, gender and additional diagnoses) [43]. At a later stage, the study also followed a chain referral sampling for pastoralists with chronic pain not treated within a healthcare facility. This technique is particularly useful to gain access to hard to reach population segments [44].

## Setting and recruitment

The study took place in primary, secondary and tertiary care facilities in the SRS as well as in pastoralist communities. We selected the study sites based on considerations of regional diversity and rural/urban differences. In doing so, we intended to gain insights into heterogeneous perceptions of chronic pain. In addition, we were interested in differences in pain severity across study sites, organizational features and care. For instance, we anticipated that we might speak to patients with more severe pain conditions in the hospital setting.

On the primary care level, we interviewed patients in two health centers in a rural area, each treating up to 25 patients per day. On the secondary care level, we recruited patients in a hospital with about 100 beds, also located in a rural area. On the tertiary care level, we recruited patients of a hospital in a more urban region. The tertiary care hospital is the largest hospital of the SRS, with more than 300 beds. We recruited pastoralist patients in these healthcare facilities with the support of the responsible health professionals. We discussed whether the current health condition of purposively selected patients allowed participation in our study.

Finally, we also interviewed pastoralists in rural communities. We chose this additional setting to reach pastoralists with chronic pain who were not receiving biomedical care. Thereby, we hoped to gain insight into possible reservations about health facilities, among other things. Pastoralist community representatives acted as gatekeepers introducing us to potential participants.

## Data collection

This study was conducted between July 2020 and January 2021. We selected semi-structured interviews for data collection to explore sociocultural aspects of pain perception, in addition to corresponding experiential and existential domains [41]. Preceding the interviews, we conducted two preliminary focus group discussions with a convenience sample of pastoralist community members. To ensure that participants felt free to express their opinions and thoughts on the topic, we separated males (n = 11) and females (n = 8) for discussion. In the focus group discussions, we addressed (among other things) the topic of chronic pain in general. In the open-ended questions, we for instance asked: "What health concerns are currently most important for you and/or your community?" or "How do you experience pain in your everyday life?" [26, p. 2–3]. In addition, we also organized a stakeholder workshop and discussed the topic together with pastoralist community representatives, health professionals and policy makers. During this preliminary project phase, we wrote down first impressions related to pain perceptions. We also drafted the interview guide inspired by Kleinman's explanatory model of illness [45]. The guide addressed five principal questions concerning, for example, etiology, pathophysiology, time, course and treatment of an illness. This approach contributed to the culture-specific understanding of a sickness episode. The findings from the focus group discussions and the workshop informed us that pain was a challenging concept to speak about [26]. We found no specific term for pain in the Somali language, as described in other research [46]. Therefore, we paid particular attention to this terminological challenge during the development of the interview guide (S2 Appendix). We piloted the guide through rigorous internal and field testing with two health professionals and two pastoralists. To encourage detailed descriptions and explanations, the questions of the semi-structured interviews were open-ended, followed by neutral probes [47]. The interview started by inquiring on general perceptions of the person's pain, followed by more specific questions concerning the impact of pain on daily life and personal pain treatment practices.

The interview language was Somali. Interviews lasted from 20 to 36 minutes. Three members of the research team conducted the interviews, including two interpreters who were fluent in Somali and familiar with the Somali pastoralist culture. For cultural reasons and due to experiences during preliminary work, female researchers conducted the interviews with female pastoralists.

Interviews in the health centers took place in a quiet location outside the building. In secondary and tertiary care hospitals, interviews took place at the patient's bedside. Within pastoralist communities, we interviewed the participants in front of their residences. After each interview the research team discussed first impressions and noted them in a research diary [36]. We audio-recorded the interviews, transcribed them word for word and anonymized them (EB, SA). A member of the research team fluent in both English and Somali translated the transcripts into English.

## Data analysis

To systematically analyze the data on a case by code basis, we applied the Framework Method according to Gale et al. [36]. This matrix-based analytic method facilitates rigorous and

transparent data management. It allowed us to perform all stages of the analysis in a systematic manner [37, 48]. The Framework Method permitted us to move back and forth between different levels of abstraction without losing sight of the original data [37]. Three researchers (EB, PvE, SA) experienced in qualitative research were involved in the data analysis. We used MAXQDA 2022 software for support.

After becoming familiar with the transcripts and field notes (stage 1), we wrote initial reflective memos and discussed them with the team in regular analysis meetings (stage 2). Inductive coding allowed us to remain open to unexpected concepts (stage 3). After coding several transcripts, we agreed on a set of codes that we applied to the other transcripts, thereby forming an analytical framework (stage 4). We clearly defined codes that we had grouped together as categories. After applying the framework to all remaining interviews (stage 5), we charted the data (including illustrative quotations) into the framework matrix–with the aim of comparing the identified themes in the text (stage 6). In the final stage (7), EB wrote four extensive analytic threads and discussed them with the research team [49]. Due to the richness of the data, we decided to focus on one analytic thread in more detail at the heart of this article.

## Context of the researchers

The first author is a PhD student experienced in applying as well as teaching qualitative research methods. She has a background in nursing and spent time working as a nurse in the Global South. Prior to conducting this research in Ethiopia, the first author was not familiar with the local context and culture. For initial familiarization, she participated in a stakeholder workshop with pastoralist community representatives and in two focus group discussions about more general health problems and needs with pastoralists in the Somali Region.

The second author is an experienced lecturer at Jigjiga University (JJU) in the Department of Health Science. He has a background in nursing and works in different research projects as a supervisor and researcher.

## Trustworthiness

To ensure trustworthiness, we followed the criteria proposed by Lincoln and Guba [50]. By triangulating data derived from the researcher's notes from the preliminary focus group discussions, stakeholder workshop, and interviews, we addressed credibility. We ensured transferability by providing detailed information on the study context, the methods and the research procedure, by ensuring a heterogenous sample in various settings and by considering rich descriptions of participants' experiences. This allows for a clustering of patterns in similar contexts and, based on this grouping, a certain generalizability. To increase dependability, we repeatedly discussed initial patterns and interpretations of the emerging codes within the multidisciplinary research team. This contributed to a high degree of reliability and consistency of the research findings. The team members established confirmability by discussing the reflective memos portraying the respondents' statements. This is a way of minimizing researchers' biases and assumptions. One co-author (PvE) supported data analysis and was not involved in the data collection process in the field. Therefore, he provided a more external perspective on the results and interpretations.

## Ethical considerations

The University of Jigjiga Ethical Review Board in the SRS approved this study (Ref. No: RERC/020/2012E.C). The Ethics Committee of Northwest and Central Switzerland (EKNZ) confirmed the fulfilment of ethical and scientific standards (Ref. No: 2020–00338). Half of interviewees were illiterate. Therefore, we also informed orally about the study and made sure a

family member was present who was able to understand the written study-related information. All interviewees signed an informed consent form prior to the interview. If the interviewee was illiterate, the consent form was signed by a fingerprint. In addition, the participant's literate representative also signed the consent form. Participation was voluntary and could be discontinued at any time. We performed this research in line with the ethical standards in the Declaration of Helsinki [51].

## Results

### Participant demographics and their pain conditions

We interviewed a total of 20 pastoralists with chronic pain. All were Muslim and except for one from the Somali ethnic group. Seven participants were agro-pastoralists, the other participants were mobile pastoralists. Most participants were married and had an average number of five children. The average number of household members was nine. Half of the participants had completed at least primary school, with the other half having no formal school training. Further sociodemographic and pain-related information is depicted in Table 1. We recruited five pastoralists (ID16 –ID20) outside the health facilities and interviewed them at their homes. At the time of the interview, participants in the hospitals and health centers reported a higher pain intensity on average than participants in the community.

Table 1 allowed us to consider the epistemological differentiation proposed by Kleinman [52]. "Disease" is conceptualized as a biological phenomenon (outsider's or etic view) and "illness" as a cultural construction (insider's or emic view). Based on this distinction, the category of "pain localization" (Table 1) represents the subjective emic perspective of pastoralists: How

**Table 1. Sociodemographic information and pain allocation.**

| | Age | Gender | Setting | Pain intensity* | Pain localization | Suspected (other) diagnoses, (potential) causes of pain |
|---|---|---|---|---|---|---|
| ID1 | 60–64 | M | Hospital[c] | 6 | Flank pain | Coronary heart failure, pulmonary edema |
| ID2 | 20–24 | M | Hospital[c] | 6 | Neck pain | Lymphoma |
| ID3 | 35–39 | F | Hospital[c] | 6 | Leg pain | Trauma: car accident |
| ID4 | 35–39 | F | Hospital[c] | 6 | No exact location | Hemiplegia, fall, cervical problems |
| ID5 | 30–34 | M | Hospital[c] | 2 | Chest pain | Community-acquired pneumonia |
| ID6 | 20–24 | M | Hospital[b] | 8 | Neck pain | Lymphoma, diabetes |
| ID7 | 60–64 | M | Hospital[b] | 4 | Leg pain | Necrotizing chronic leg wound |
| ID8 | 55–59 | M | Hospital[b] | 2 | Backpain, headache | Malaria, heart problems |
| ID9 | 30–34 | F | Hospital[b] | 6 | Stomach pain | Peptic ulcer, dyspepsia |
| ID10 | 35–39 | F | Hospital[b] | 4 | Backpain, headache | Neuropathic fever, pancreas problems |
| ID11 | 20–24 | M | Health Center[a] | 4 | Leg pain, headache | Malaria |
| ID12 | 30–34 | F | Health Center[a] | 6 | Headache | Malaria |
| ID13 | 35–39 | M | Health Center[a] | 4 | Abdominal pain | Urinary tract infection |
| ID14 | 35–39 | F | Health Center[a] | 4 | Backpain, headache | Malaria |
| ID15 | 30–34 | F | Health Center[a] | 8 | Chest pain | Tuberculosis |
| ID16 | 45–49 | M | Community | 2 | Headache | Skin cancer |
| ID17 | 45–49 | M | Community | 0 | Backpain | None stated |
| ID18 | 40–44 | F | Community | 4 | Stomach pain | Typhoid, gastritis |
| ID19 | 25–29 | M | Community | 4 | Headache, leg pain | Trauma: violence |
| ID20 | 35–39 | F | Community | 2 | Backpain, headache | Malaria, female genital cutting |

*At the time of the interview; according to the self-report measure Faces Pain Scale–Revised (FPS-R) from 0–10 a) Primary care level b) Secondary care level c) Tertiary care level

did they themselves describe their pain and how did they understand their lived illness experience? By contrast, the category "suspected (other) diagnoses, (potential) causes of pain" provided information on the objective, scientific etic perspective held by health professionals. This category was based on the diagnosis provided in the health facility and thus represents the biomedical diagnosis.

## Pastoralists' perceptions of pain

Our analysis suggests that participants' perceptions of their chronic pain can be clustered into six different themes: "Pain as a symptom of harsh daily life", "Pain descriptions and dimensions", "Temporality of pain", "Stigma and stoicism", "Mediating role of spirituality"; and "Impact of pain on daily life activities". These themes are related to the main domains of the biopsychosocial model [40] and the approach of Stilwell and Harman (Fig 1) [14].

We deliberately avoided a clear attribution of chronic pain perception to one specific domain since this would lead to a significant loss of notion and experience. Hence, the model represents the overlap of domains and is intended to represent the fluid and dynamic nature of pain perception.

## Pain as a symptom of harsh daily life

Most participants and particularly pastoralists from rural areas described pain as something normal everyone must face. Pain was part of their daily life and, therefore, something they had learned to live with. The majority of interviewees described loading animals, harvesting or fetching and carrying water over long distances as strenuous activities that were part of their daily routines. However, they rarely described such daily activities as the sole cause of their

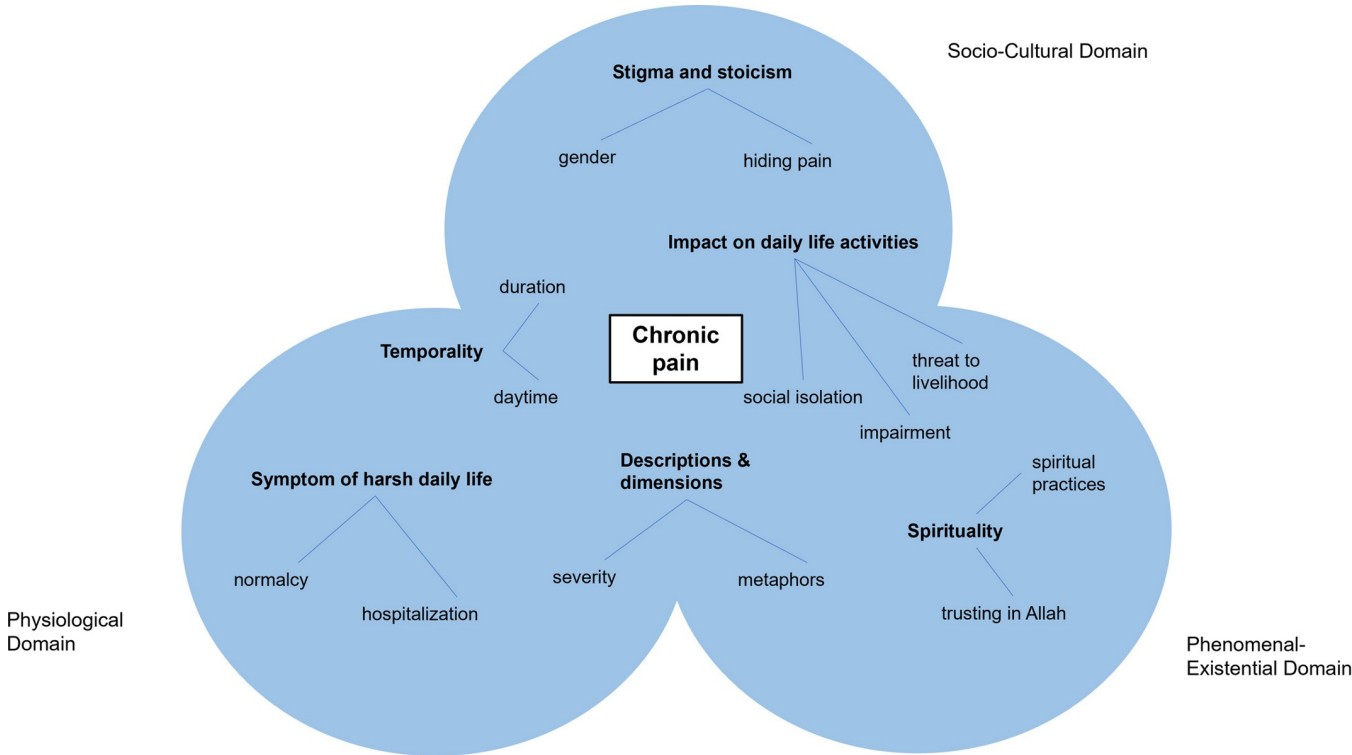

**Fig 1. Chronic pain perception of Somali pastoralists.** Inspired by Coninx and Stilwell [41] and de Haan [53].

pain. A few participants mentioned that they had experienced female genital cutting (FGC) themselves or knew someone in their community who had undergone FGC. Yet they did not bring up that they currently suffered from pain due to FGC.

At first, the interviewees often described pain as an acute event related either to an injury from an accident or fight or to childbirth. By applying more in-depth probing questions, we could also reveal their perceptions of chronic pain. In rural areas, such as at the primary care level or in the community, pastoralists most commonly associated their chronic pain with a communicable disease such as malaria, typhus or tuberculosis. This became evident when interviewees spoke of being sick in general or having a fever. This was not surprising due to the connotation of the word (i.e., the Somali word for "pain" meaning both pain and illness). As an example, a woman in a rural area referred to the harsh conditions she and the members of her community live in:

> *"In our entire life, everybody will experience pain, you might suffer from labor pain. [. . .] We may feel headache, even you may feel headache in the evening if you work and walk during the day. There also may be malaria. But we receive medication for that. I have not suffered from headache. But I did get malaria. So, I got treatment for it within a short time. Everyone who stays here might get malaria." (ID20, line 16)*

Participants described pain as a distinctly relational phenomenon, a bodily sensation that rarely existed on its own. They associated it as a symptom of some other form of physical ailment. Interestingly, one participant stated that the health professionals were the ones who gave pain a particular name. This allowed the participant to make sense of his pain. Interviewees perceived chronic pain as something that often changed its intensity over time, for instance, depending on the sort of activity they had to undertake or depending on the time of the day.

In secondary and tertiary health facilities, however, we found that patients described their pain as no longer "bearable" in daily life. They had tried to "stay" with the pain as long as possible at home before taking on the often very long and costly journey to the next hospital. The pastoralists decided to come to the hospital when they were no longer able to cope with their pain. Commonly, they described this experience as no longer being able to conduct the chores expected of them.

### Pain descriptions and dimensions

To communicate the pain experience, pastoralists often used figurative language, metaphors or anthropomorphisms. They compared their pain with something that resembles it–in order to create understanding. Metaphors and anthropomorphisms served as a means to convey the quality of the pain, its duration and association. Most participants mentioned that their pain was very severe when they had decided to seek care. At the time of the interview, most participants stated having moderate pain. To illustrate the severity of pain and its consequences, an elderly man used a figurative language:

> *"People were recommending me to stay and wait until it disappears on its own and then I felt severe pain. Even my hair became grey last year from the tumor due to the pain." (ID16, line 17)*

While talking to us, the man pointed at this hair. With this gesture, he directed our attention to the obvious consequences of severe pain–his white hair as a visible proof of very intensive pain. Another participant mentioned the self-treatment for pain by burning his own skin. When this treatment did not help, he described the anthropomorphism of "haunting" pain–as if pain were a human being persecuting the pain-affected person. Pain *"breaking legs and toes"*

or *"sucking a person's blood"* were further anthropomorphisms. Moreover, anthropomorphisms also served to indicate the effect of medication. A patient mentioned that the drug he received restituted his mobility:

> *"At first, I was in pain for a long time but it was not so severe and I just kept going so I used to take a pill called in local language "Habartosiye" which means a pill that makes old women stand up." (ID7, line 17)*

A female patient compared her pain experience with the feeling of her stomach "being cut". She questioned the efficacy of the medication received for treatment. Her pain continued to be so unbearable at times that the only possible response was "to run away:

> *"Occasionally, I run away from my home due to the severity of the pain. I could not bear it. [. . .] It is due to the severity of the pain, which is from gastritis. Sometimes I feel like my stomach is being cut. And maybe the medication is not the right one." (ID9, line 13)*

Participants clearly stated the importance of medication to treat their pain. In health facilities, they expected to receive a pharmacologic therapy. However, several of the interviewees reported that they did not receive the necessary pain medication. This was especially the case in primary care. The lack of pain relief resulted in doubts about the right therapy. Patients' non-compliance and discontinuation of medication were the consequences.

Some participants clearly attributed the pain to a specific location in the body. However, for others describing the pain location was more difficult. A female participant with several chronic conditions stated:

> *"My pain is generalized to most parts of my body. I cannot locate the exact place." (ID4, line 16)*

In addition, several participants used gestures, pointed to the pain location on their body or touched the painful area. Even when they were not able to determine the location or the direct cause, they perceived pain as a bodily issue and not as a psychological or psychosomatic problem. However, some pastoralists mentioned that their physical pain triggered negative emotions like sadness, anxiety or aggression.

## Temporality of pain

Participants reported very different pain durations ranging from a couple of months to several years. Particularly, community members in rural areas mentioned having chronic pain for a very long time, even since childhood or adolescence. Although we focused on chronic pain in this study, some participants also reported acute painful conditions they experienced at the same time:

> *"I have been suffering from gastritis for around one year. Eating different meals aggravates the problem but currently I have been suffering from backpain and from a cough for fifteen days. Moreover, I have a headache, which is almost continuous. Now the pain I feel worst is backpain." (ID8, line 12)*

Another temporal aspect of chronic pain perception was the time of highest pain intensity. The majority of participants mentioned that their pain felt worse during the night. According to some interviewees, this was due to cold temperatures at night. One younger woman in primary care explained this assumption in the following way:

*"Since it is cold at night, most of the body works less because the blood gets cold. Therefore, when you try to sleep, you feel more pain. But during the day, it is different because your body is hot and it works more." (ID15, line 45)*

Another young male participant in secondary care described his experience in a similar manner:

Q: *"Why do you think pain is worse at night?"*

A: *"I don't know why it is worse at night. Maybe it is due to the cold of the night." (ID6, line 43)*

Several participants mentioned their "blood becoming cold" at night. They related this to higher pain intensity. Only very few participants felt that the intensity of their pain was the same throughout the day and in the night.

## Stigma and stoicism

In both genders, stoicism with regard to pain expression and reporting of pain was an issue that was often touched upon. Particularly in rural areas, stoicism or impassiveness was a topic–associated with age and marital status. Younger, unmarried adults and especially women were more inclined to bear or hide their pain, since they were more worried about stigma:

*"Young unmarried women hide their pain but married women show their pain to their husband. [. . .] She is afraid of being stigmatized, and also men may not come close to her due to her health condition." (ID17, line 58)*

In one interview, a woman mentioned hiding her pain to not worry other persons. She associated this attitude with her Muslim religion:

*"The reason why we don't describe our pain early when the pain is minor is–since we are a Muslim society–it is not good to terrify or scare people to worry them by complaining more about some minor pains." (ID15, line 36)*

Another female participant also described bearing her pain when it is less severe:

*"It depends on the severity of the pain. If the pain is very severe and if I am able to communicate with the people, I can report the pain I am feeling. But if the pain is mild or minor, I will try to bear the pain, keep it to myself." (ID9, line 40)*

Similarly, an older male participant also stated that young men might not show their pain out of fear of social stigma:

*"Young men also hide their pain, because they are afraid of being called a coward and maybe stigmatized while old men show their pain." (ID17, line 56)*

However, several men were of the opinion that women are particularly vulnerable towards pain because they "hide" or bear it and seek care at a very late stage. A male participant from a rural area, for instance, made the following statement:

*"Men can bear the pain compared to women but women hide their pain. Women are being taken to the health center. They deliver there. So, they dislike that male health providers may observe or administer injections so they hide their pain. [. . .] Currently, women get broken teeth frequently, because they hold their teeth tightly to stand the pain. Women frequently get sick, because when they get sick, they don't seek health care." (ID16, 53)*

Pastoralist women themselves did not explicitly mention this kind of pain vulnerability. Some women, however, did describe uneasiness with regard to male health providers and difficulty reporting their pain to others. Several men mentioned that women felt ashamed or feared further health interventions or diagnostics if they informed others about their pain. From their point of view this may result in additional check-ups performed by male health providers.

## Mediating role of spirituality

The vast majority of pastoralists expressed deeply religious feelings. Their statements revealed that spirituality played a crucial part in the perception of their pain. According to their religious conviction, pastoralists from rural areas mentioned that their destiny is in the hands of Allah:

*"Allah is the one who causes the pain, everyone will get it." (ID20, line 31)*

Participants mentioned that they trusted Allah to know the source of their pain. For example, an elderly man said:

*"I don't know why I am suffering from this pain; Allah may know." (ID8, line 64)*

Similarly, a younger woman answered the question "Why do you think you are suffering from this pain?" in the following way:

*"Allah may know, I took a lot of medications." (ID9, line 17)*

Another woman emphasized that her fate is in Allah's hands:

*"My illnesses will be either managed or not managed. Thanks to Allah." (ID18, line 9)*

In rural primary care as well as in the community setting, dealing with pain was strongly associated with spiritual practices. Participants mentioned praying and reciting the Koran as the most important initial care practices to improve one's health or the health of other sufferers. One participant described her faith and care seeking behavior in the following way:

*"Religion is the backbone of our life. Whenever somebody gets sick, we first recite the Koran and then we may go to the health center." (ID18, line 27)*

For pastoralist patients, practicing Islam was a way to deal with their pain in an early stage, commonly followed by home remedies/traditional medicine or visits to the traditional healer– before finally seeking care in a biomedical health facility.

## Impact on daily life activities

On the one hand, participants emphasized that pain was part of their daily life. On the other hand, they also described the negative impact of chronic pain on their daily lives. Most

participants mentioned the consequences of severe pain on their work. They were no longer able to carry out their chores as required due to their impaired mobility. Several male participants mentioned that taking care of their livestock and sustaining the family became difficult due to their chronic pain. This could finally lead to feelings of hopelessness:

> *"It [the pain] affected me more even I could not keep my sheep and one of the sheep got a leg fracture [i.e., the animals were neglected]. And I have six children and I couldn't do anything for my children. That was the impact of the pain on daily life." (ID3, line 56)*

Women reported no longer being able to clean and cook or feed their children. One participant's husband described that men are affected when women are in pain:

> *"When the woman gets sick, is suffering from pain–the whole home is affected and everyone in the home feels it. Most of the home activities are not done well." (ID1, line 89)*

Interviewees described becoming more and more dependent on other family members. They also mentioned financial difficulties and precarious situations due to loss of income:

> *"The pain had a great impact on my wealth, my life and my family, slaughtering and selling different sheep or goats, visiting different health facilities, and also I lost weight. It was hard for me to work." (ID17, line 81)*

Moreover, some participants reported that families were forced to sell their animals to pay for treatment:

> *"Currently we only own few livestock, we used to have more, we sold some of our animals to pay for her treatment and some others died due to the drought." (ID9, line 55)*

In this context, a participant described his situation as follows:

> *"I used to own more animals, but they became less. Some got lost because there was no one who could tend to them, while some others covered my expenses for treatment." (ID17, line 93)*

The majority of participants mentioned their threatened position within the community as a consequence of chronic pain. They were no longer able to engage in social and religious activities. Social isolation or marginalization became evident in the interviews when several participants expressed their wish to keep to themselves when in severe pain:

> *"I don't like to stay with my friends because I can't talk with them. They want me to converse with them like before so I can't do that and I prefer to stay alone and take rest." (ID2, line 40)*

However, spending time with others and seeking distraction from pain was also a coping mechanism when pain was less severe:

> *"If the intensity of the pain is increasing, the person will be in bed even if the person is male. Now I feel better while I am conversing with you but when you leave the pain will relapse." (ID1, line 93)*

## Discussion

This study explored Somali pastoralists' perception and notions of their chronic pain. To our knowledge, this is the first study exclusively addressing this topic among pastoralists in the SRS. Our findings highlight the complex und multicausal experience of chronic pain among members of this marginalized population. In the following we discuss the main findings through the lens of the proposed enactive framework [14].

Primarily, the participants described pain as something corporeal related to the physiological domain. This is in line with the findings of other studies with Somali patients [46]. While chronic pain can be defined as an empirical disease in itself [1], pastoralists explained their pain as a symptom of an underlying physical disease or tangible health problem and not in the light of a psychological problem. However, this is not to say that Somali pastoralists do not experience psychological pain. On the contrary, the interviewees often had experienced serious existential loss and drastic hardship most certainly leaving psychological and emotional scars (e.g., devastating droughts, incisive loss of livelihoods). Interestingly, Schuchman and McDonald [54] argue that Somalis may express emotional suffering through bodily complaints, for instance, by reporting headache or sleeplessness. Finnström and Söderhamn [32] also argue that Somali women with pain in some cases report their pain through "the medium of the body" (p. 421) since they are unaware of the pain's psychosomatic nature.

In consideration of the socio-cultural domain, these findings can also be discussed in the light of Kleinman's concept of illness and disease [52] as portrayed in Table 1. According to Kleinman, the social responses to pain refer to the subjective illness experience. The person's pain (illness) is thereby deeply shaped by culture-specific contexts or environments (emic perspective), as indicated in the enactive framework [14]. In contrast, the concept of disease is objective and empirically evident. The physical abnormality is measurable by means of different diagnostic tests and identified by a health professional (etic perspective) [45]. In this study we found that the biomedical term for the health problem provided by the health professional turned the illness into a (socially) recognized and accepted (often communicable) disease for the pastoralists. Pain became a relational and compatible symptom of that particular disease [14]. However, in line with other research, pastoralists' descriptions oscillated between their pain experience and other illness symptoms. This may be a consequence of the ambiguous term "xanuun" referring to both pain and illness [46].

At times, we also had the impression that the provided etic diagnosis (e.g., an acute infectious disease) did not always clearly correspond with the illness experience described by a pastoralist patient. This could lead to other symptoms more chronic in nature from being overlooked or underrated (e.g., persisting backache).

Our findings confirm that social responses to pain within the pastoralist's environment shape the way pain is perceived, accepted and dealt with [14, 55]. Negative reactions of peers invalidating the person's pain may cause pain to be less accepted and increase the risk of experiencing persistent pain [56, 57]. Pastoralists in our study found a way to report their pain in a manner that was culturally accepted in their community. They described the phenomenon by reporting an underlying illness or somatic abnormality, as opposed to indicating psychological causes. Results reported by Finnström and Söderhamn [32] as well as by Campeau [46] indicate that seeking psychological help is less accepted in Somali culture. We also found that pastoralist patients rarely communicated minor pains because of feared undesired social or legal repercussions. Gender concerns regarding male health professionals were also relevant. Although FGC is still highly prevalent in the SRS [58], very few women mentioned that they suffered from FGC or knew someone who had experienced FGC and associated chronic pain. Referring to Foucault's concept of biopower and power-knowledge [59], we found that the

regional government had cracked down on this practice by strictly banning it. Consequently, women were afraid to speak about it. Discourses on FGC were manipulated out of fear of surveillance and punishment. Research with Somali patients living in other contexts has also found that women were very hesitant to talk about sensitive problems, specifically about female circumcision [32]. Not mentioning these issues could additionally intensify the women's self-isolation.

Our findings also confirm other studies indicating a general stoic attitude towards pain in the Somali culture [25] and among other pastoralist people populations [60, 61]. This mental stance together with other sociological factors held pain-affected persons back from help-seeking and resulted in underreporting of pain. In line with descriptions by Zinsstag et al., the interviewed pastoralists sought biomedical care at a very late stage [33]. Only upon request, the interviewees mentioned hard physical labor, that they themselves did not necessarily consider to be strenuous, and only rarely did they link their chronic pain with such exhausting activities and their daily chores.

Our findings indicate that particularly women did not report or tried to bear their pain due to fear of imminent stigmatization. Such expected negative socio-cultural signals can limit the ability to act and in fact increase pain [14]. This especially becomes relevant for pastoralists, who felt that social commitment in fact helped ease their pain. The fear of stigmatization could result in less social engagement and therefore, even intensify pain. This is important to consider as socio-cultural stigma in light of chronic illness can have a severely negative impact on a person's health with an increased risk of morbidity and mortality [62].

Another important finding referred to pastoralists' use of idioms, metaphors and anthropomorphisms to describe their corporal pain and its intensity ("pain is sucking a person's blood"). This is another example for the pervasiveness of the socio-physiological domain. In this regard, other studies also reported the metaphorical language as a means of communicating one's pain and of making others understand it [63]. According to Lakoff and Johnson [64], metaphors are important tools for health professionals to better understand their patients' emotions, cultural practices, and spiritual beliefs associated with their illness experience.

One of the central results concerning the socio-cultural and phenomenal-existential domains was the significance of spirituality for coping with chronic pain. The majority of pastoralists considered their religious faith to be interconnected with the etiology of their pain. Not surprisingly, a central coping mechanism for pastoralists to deal with their pain was to pray and recite the Koran. These findings complement earlier research with Somali patients, stating that the Muslim faith provided its followers with meaningful answers to their suffering and to help alleviate it [46, 65]. Evidence from international studies further emphasizes that spirituality can increase a person's acceptance of his or her pain [66, 67]. However, these religious beliefs and their coping practices did not contradict and thus inhibit genuine biomedical treatments. The biomedicalization and arguably the "pharmaceuticalization" of chronic pain were processes indirectly hinted at during several interviews. Pastoralists assessed this critically (i.e., they did not trust the efficacy of the treatment) and simultaneously felt attracted to it (i.e., they highly esteemed effective pharmaceuticals).

Our findings also highlight that chronic pain had a detrimental effect on pastoralists and on their families. It permeated all aspects of their daily life and resulted in financial hardship, loss of roles in the family or community and social isolation. The enactive framework can offer several possibilities for interpreting the culturally shaped perception towards chronic pain: The lack of treatment options and prospects for improvement can cause feelings of despair and hopelessness. This resulted in pastoralists with chronic pain having difficulties perceiving attractive future affordances [14]. Social isolation due to chronic pain proves to be even more detrimental in a society characterized by strong social ties [18].

## Strengths and limitations

In this study we interviewed a heterogeneous purposive sample of pastoralists with different chronic pain conditions. By applying ethnographic research techniques, we were able to gain a broader and more in-depth understanding of pastoralist patients' chronic pain perceptions. By considering various settings, including urban and rural contexts as well as primary, secondary and tertiary levels of care, we were able to draw interesting comparisons. Importantly, we also interviewed pastoralists at their homes.

Our multidisciplinary research team allowed complementary, mutually enriching perspectives and, therefore, added to overall trustworthiness of the findings. Furthermore, our preliminary focus group discussions with 19 additional pastoralists increased the credibility of our findings.

A limitation of this study refers to a potential social desirability bias. Some participants might have described a more positive experience with health services in order to be in accordance with the researchers' opinion. We minimized such bias by applying different strategies recommended by Bergen and Labonté [68]. For example, we stated that the expression of one's personal opinion will not have any consequences. Furthermore, we tried to create a trusting environment with the interviewees by showing interest, empathy and respect. We also ensured confidentiality and a private atmosphere without the presence of a third person in most interviews. In the hospital setting it was not always possible to talk to women in private. Their husbands or other family members often remained at their bedside. Yet, we also learned that family members provided additional insights. Their presence sometimes motivated patients to share their story with us.

Another limitation refers to the lack of any direct observations concerning participants' daily lives on a household level. Moreover, it was not possible to conduct additional interviews or to repeat interviews in order to fully establish data saturation of all themes due to political unrest and drought in the region during our field study. Finally, the sample size of 20 purposively selected pastoralists calls for recommendations to be considered with caution. Nevertheless, the qualitative findings allow a more in-depth understanding of pastoralists' perceptions of chronic pain. They provide novel insights into their illness experience in diverse settings. The results contribute valuable knowledge on this marginalized population and can guide future studies with a larger sample size.

## Conclusion

Pastoralists perceived their chronic pain to be a multicausal and multifaceted relational experience. The pastoralists' pain was part of their harsh daily life and therefore a physical symptom of prevailing deprivation and experienced hardships. Thus, pastoralists had to find ways to deal with their pain within their closest circle. They only sought professional care when their pain became severe. This calls for more research investigating how pastoralists cope with the burden of chronic pain in their homes and communities.

Receiving a biomedically recognized and socially more acceptable somatic diagnosis made the pain and their individual illness experience more tangible and potentially easier to cope with. The pastoralists' reservation towards communicating their pain and seeking biomedical care can be linked to aspects of social stigma and stoicism concerning pain expression, especially among women. The use of metaphors and anthropomorphisms to describe pain quality, intensity and temporality plays a significant role. By sharing individual pain experiences with others, pastoralists attempted to achieve a common understanding held by family members and health professionals. The mediating role of spirituality supported pastoralists to give meaning to their pain and to relieve its debilitating impact on their daily life.

The findings of this study can foster a more in-depth understanding of pastoralist patients' perception of chronic pain. Since there is no specific term for pain in the Somali language and pain can have multiple etiologies, the phenomenon requires in-depth exploration in order to fully understand it. Therefore, health professionals must have outstanding communication skills and knowledge on pain management in this context to adequately assess and treat pastoralists' pain. This implies deriving appropriate practices addressing not only the physiological domain but also psychosocial concerns, existential issues and spiritual beliefs. The social stigma concerning the expression of pain, especially for female pastoralists in rural areas, indicates that accessibility to comprehensive and culturally sensitive pain treatment is a problem for pastoralists. Therefore, female health professionals who are also trusted members of the community and trained in pain management may assume a key role. They could visit female patients instead of seeing them in the health facilities.

Pastoralists have a citizen right to request and receive adequate treatment for their pain. According to international human rights laws, government authorities are obligated to ensure that their citizens have access to treatment for their pain. In this sense, pain is considered a serious public health issue [69]. Therefore, government authorities have a responsibility to prioritize the improvement of pain management on all levels in the SRS. They should fulfill their obligation to ensure available, accessible, affordable, and culturally acceptable services for pastoralist patients.

Finally, existing research on chronic pain is significantly influenced by a Western-shaped, biomedically informed understanding of health in general and of pain in particular. Hence, our findings support the pressing demand for more inclusive and decentered pain research in Sub-Saharan Africa, especially with regard to marginalized communities such as pastoralists.

## Supporting information

**S1 Appendix. Consolidated criteria for reporting qualitative research (COREQ).**
(DOCX)

**S2 Appendix. Semi-structured interview guide.**
(DOCX)

## Author Contributions

**Conceptualization:** Eleonore Baum, Nicole Probst-Hensch, Jakob Zinsstag, Birgit Vosseler, Peter van Eeuwijk.

**Data curation:** Eleonore Baum.

**Formal analysis:** Eleonore Baum, Sied Abdi, Peter van Eeuwijk.

**Funding acquisition:** Jakob Zinsstag.

**Investigation:** Eleonore Baum, Sied Abdi, Rea Tschopp.

**Methodology:** Eleonore Baum, Peter van Eeuwijk.

**Project administration:** Rea Tschopp.

**Supervision:** Nicole Probst-Hensch, Jakob Zinsstag, Birgit Vosseler, Peter van Eeuwijk.

**Validation:** Sied Abdi, Rea Tschopp.

**Writing – original draft:** Eleonore Baum, Peter van Eeuwijk.

**Writing – review & editing:** Sied Abdi, Nicole Probst-Hensch, Jakob Zinsstag, Birgit Vosseler, Rea Tschopp.

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
