## [Decision Letter · Decision Letter 0]

6 Sep 2023

PONE-D-23-08854“I could not bear it”: Perceptions of chronic pain among Somali pastoralists in Ethiopia. A qualitative studyPLOS ONE

Dear Dr. Baum,

Thank you for submitting your manuscript to PLOS ONE. After careful consideration, we feel that it has merit but does not fully meet PLOS ONE’s publication criteria as it currently stands. Therefore, we invite you to submit a revised version of the manuscript that addresses the points raised during the review process.

We look forward to receiving your revised manuscript.

Kind regards,

Taofiki Ajao Sunmonu

Academic Editor

PLOS ONE

Journal Requirements:

a) Did participants provide their written or verbal informed consent to participate in this study?

“This 10-year project is part of the Jigjiga University One Health Initiative (JOHI) co-funded by the Swiss Agency for Development and Cooperation (SDC) Project no. 7F-09057.02.01. The project is implemented by Jigjiga University (JJU), Swiss Tropical and Public Health Institute (Swiss TPH) and the Armauer Hansen Research Institute (AHRI). The project is also supported through the doctoral program at OST - Eastern Switzerland University of Applied Sciences and the scholarship program from the Swiss School of Public Health (SSPH+). The funders had no role in study design, data collection and analysis, decision to publish, or preparation of the manuscript.”

Additional Editor Comments (if provided):

it would be nice if the sample size of the study could increaed as suggested by the reviewers.

Reviewers' comments:

Reviewer's Responses to Questions

**Comments to the Author**

1. Is the manuscript technically sound, and do the data support the conclusions?

Reviewer #1: No

Reviewer #2: Yes

2. Has the statistical analysis been performed appropriately and rigorously? 

Reviewer #1: Yes

Reviewer #2: N/A

3. Have the authors made all data underlying the findings in their manuscript fully available?

Reviewer #1: No

Reviewer #2: Yes

4. Is the manuscript presented in an intelligible fashion and written in standard English?

Reviewer #1: Yes

Reviewer #2: Yes

5. Review Comments to the Author

Reviewer #1: Although, this study aims to explore the perceptions and notions of chronic pain among Somali pastoralists but I think this illustration cannot add more to the scientific research and the authors better use a control group from other country or another population to strenghthen the study.

Reviewer #2: I reviewed a manuscript titled "Perceptions of Chronic Pain Among Somali Pastoralists in Ethiopia." This is a small descriptive study that looked at a diverse group of adult pastoralists and agro-pastoralists dealing with chronic pain in Ethiopia. The study used ethnographic research techniques. Pastoralists are a group often marginalized socially and geographically, known for enduring many hardships in their daily lives. When it comes to chronic pain, they mainly rely on traditional and spiritual health practices, with Western biomedical approaches being quite rare. The authors referenced a study by Kawza et al., which found that only 10% of pastoralists in Southern Ethiopia used biomedical healthcare facilities when they were sick.

Main strength of this study is how it enhances our understanding of the sociocultural aspects of pain perception within this isolated population.

However, a major weakness lies in the small sample size and the potential bias towards patients who use biomedical healthcare facilities.

6. PLOS authors have the option to publish the peer review history of their article (what does this mean?). If published, this will include your full peer review and any attached files.

Reviewer #1: No

Reviewer #2: No

---

## [Author Response · Author response to Decision Letter 0]

30 Sep 2023

RESPONSE TO REVIEWERS

Manuscript: PONE-D-23-08854

Manuscript title: “I could not bear it”: Perceptions of chronic pain among Somali pastoralists in Ethiopia. A qualitative study

Comments from the Editor 

It would be nice if the sample size of the study could increased as suggested by the reviewers.

Response from the Author

Thank you for your suggestion. We are convinced that qualitative studies with smaller sample sizes can provide valuable insights into a marginalized population that so far has been neglected in this field of research. We have now added more information on the sample size of the preceding focus group discussions in the methods section. We also mentioned this as a major strength of our study. Thereby, we can increase the overall sample size of the study. 

- Methods section, p. 7, lines 181-186

- Strengths and limitations section, p. 23, lines 572-575

Comments from Reviewer 1

Although, this study aims to explore the perceptions and notions of chronic pain among Somali pastoralists but I think this illustration cannot add more to the scientific research and the authors better use a control group from other country or another population to strenghthen the study. 

Response from the Author

Thank you for this comment. We argue that qualitative research has a great deal to offer when trying to gain in-depth insights into persons' illness experiences. From our point of view, the findings are completely novel in this field of research. To our knowledge no other study has examined chronic pain experience in this unique context and in this marginalized population. Indeed, these initial findings can provide valuable insights and orientation for future studies (as described in the "strengths and limitations"). In addition, by drawing on studies investigating Somalis and/or pastoralists in other countries, we can make relevant comparisons (e.g., lines 483-493). Furthermore, we made sure to now point out that the various settings of this study also allowed for interesting comparisons (home as well as biomedical). 

- Methods section, p. 7, lines 181-186

- Strengths and limitations, p. 23, lines 570-575/590-593

Comments from Reviewer 2

I reviewed a manuscript titled "Perceptions of Chronic Pain Among Somali Pastoralists in Ethiopia." This is a small descriptive study that looked at a diverse group of adult pastoralists and agro-pastoralists dealing with chronic pain in Ethiopia. The study used ethnographic research techniques. Pastoralists are a group often marginalized socially and geographically, known for enduring many hardships in their daily lives. When it comes to chronic pain, they mainly rely on traditional and spiritual health practices, with Western biomedical approaches being quite rare. The authors referenced a study by Kawza et al., which found that only 10% of pastoralists in Southern Ethiopia used biomedical healthcare facilities when they were sick.

Main strength of this study is how it enhances our understanding of the sociocultural aspects of pain perception within this isolated population.

However, a major weakness lies in the small sample size and the potential bias towards patients who use biomedical healthcare facilities. 

Response from the Author

Thank you for your considerations and for highlighting these important points. We added further information on the utilization of health services specifically for Somali pastoralists. 

As elaborated briefly above, we are convinced that qualitative studies with small sample sizes can provide valuable insights into a population, for which we have very limited information on pain perception. The small sample size was mentioned in the "strengths and limitations" section of the paper. We have now added more information on the sample size of the preceding focus group discussions. Indeed, most interviews took place in biomedical healthcare facilities. However, we also interviewed n=5 pastoralists affected by chronic pain in their home. We elaborate on this in the section "Setting and recruitment". In addition, we highlight the pastoralists' care itineraries in the results, thereby describing steps taken before they came to the health facility. In the "Conclusion", we added the recommendation to investigate how pastoralists are burdened by chronic pain and how they cope with chronic pain within their communities. 

- Background section, p. 4, lines 88-94/99-103

- Methods section, p.7, lines 181-186

- Results section, p. 10, lines 268-269

- Strengths and limitations, p. 23, lines 572-575/591-593

- Conclusion section, p. 24, lines 598-600

---

## [Editor Report · Decision Letter 1]

6 Oct 2023

“I could not bear it”: Perceptions of chronic pain among Somali pastoralists in Ethiopia. A qualitative study

PONE-D-23-08854R1

Dear Dr. Baum,

We’re pleased to inform you that your manuscript has been judged scientifically suitable for publication and will be formally accepted for publication once it meets all outstanding technical requirements.

Kind regards,

Taofiki Ajao Sunmonu

Academic Editor

PLOS ONE

Additional Editor Comments (optional):

The authors have satisfied the concerns raised in the article and it is pubblishable in this present form. Congrattulations.
---

## [Editor Report · Acceptance letter]

3 Nov 2023

PONE-D-23-08854R1 

“I could not bear it”: Perceptions of chronic pain among Somali pastoralists in Ethiopia. A qualitative study 

Dear Dr. Baum:

I'm pleased to inform you that your manuscript has been deemed suitable for publication in PLOS ONE. Congratulations! Your manuscript is now with our production department. 

Kind regards, 

on behalf of

Dr. Taofiki Ajao Sunmonu 

Academic Editor

PLOS ONE